# The Natural Alkaloid Palmatine Selectively Induces Mitophagy and Restores Mitochondrial Function in an Alzheimer’s Disease Mouse Model

**DOI:** 10.3390/ijms242216542

**Published:** 2023-11-20

**Authors:** Da-Ye Lee, Kang-Min Lee, Jee-Hyun Um, Young-Yeon Kim, Dong-Hyun Kim, Jeanho Yun

**Affiliations:** 1Department of Biochemistry, College of Medicine, Dong-A University, Busan 49201, Republic of Koreakmlee8348@dau.ac.kr (K.-M.L.); umj1127@dau.ac.kr (J.-H.U.); kyy8191@dau.ac.kr (Y.-Y.K.); 2Department of Translational Biomedical Sciences, Graduate School of Dong-A University, Busan 49201, Republic of Korea; 3Department of Pharmacology and Department of Advanced Translational Medicine, School of Medicine, Konkuk University, Seoul 05029, Republic of Korea; mose79@kku.ac.kr

**Keywords:** palmatine, mitophagy, mitochondrial dysfunction, Alzheimer’s disease

## Abstract

Palmatine, a natural alkaloid found in various plants, has been reported to have diverse pharmacological and biological effects, including anti-inflammatory, antioxidant, and cardiovascular effects. However, the role of palmatine in mitophagy, a fundamental process crucial for maintaining mitochondrial function, remains elusive. In this study, we found that palmatine efficiently induces mitophagy in various human cell lines. Palmatine specifically induces mitophagy and subsequently stimulates mitochondrial biogenesis. Palmatine did not interfere with mitochondrial function, similar to CCCP, suggesting that palmatine is not toxic to mitochondria. Importantly, palmatine treatment alleviated mitochondrial dysfunction in PINK1-knockout MEFs. Moreover, the administration of palmatine resulted in significant improvements in cognitive function and restored mitochondrial function in an Alzheimer’s disease mouse model. This study identifies palmatine as a novel inducer of selective mitophagy. Our results suggest that palmatine-mediated mitophagy induction could be a potential strategy for Alzheimer’s disease treatment and that natural alkaloids are potential sources of mitophagy inducers.

## 1. Introduction

Mitochondrial dysfunction has been implicated in various human diseases, especially in neurodegenerative diseases, including Alzheimer’s disease (AD), Parkinson’s disease (PD), Huntington’s disease (HD), and amyotrophic lateral sclerosis (ALS) [1]. While various strategies have been attempted to enhance mitochondrial function [2], mitophagy stimulation has been proposed as a promising strategy for reversing mitochondrial dysfunction [3,4]. Through selectively removing damaged mitochondria using core autophagy machinery and simultaneous stimulation of mitochondrial biogenesis, mitophagy plays a pivotal role in maintaining mitochondrial homeostasis [4]. Recent extensive studies have revealed various mitophagy pathways [5,6]. In addition to the PINK1-Parkin pathway, the most well-characterized pathway, various pathways, such as receptor-mediated mitophagy pathways and the ULK1-Rab9 alternative pathway, have been shown to mediate mitophagy induction in various circumstances [7,8,9]. Although mitophagy induction has been shown to have therapeutic potential in various neurodegenerative disease models, clinically applicable mitophagy inducers are very limited [10,11]. Most widely used mitophagy inducers, such as carbonyl cyanide m-chlorophenyl hydrazone (CCCP) and oligomycin/antimycin, are not suitable for clinical application due to their toxicity to mitochondrial function and cellular viability [4]. Thus, the identification of mitophagy inducers with low toxicity is essential for mitophagy-based disease treatment.

Palmatine is a natural alkaloid derived from a variety of plant sources and has been shown to possess many biological activities, including anti-inflammatory and antioxidant activities [12,13]. Previous studies have shown that palmatine has protective effects on the liver and kidney and antidiabetic effects in various in vitro and in vivo models [14]. In addition, cardioprotective [15], neuroprotective [16], and antibacterial functions of palmatine have also been reported [17,18]. Although previous studies have shown that palmatine has various physiological effects, its role in mitochondrial function and mitophagy has not been investigated.

In this study, we explored the function of palmatine in mitophagy regulation. We found that palmatine efficiently induces cellular mitophagy activity but exhibits low toxicity to mitochondria, unlike CCCP. Importantly, palmatine treatment reversed mitochondrial dysfunction in PINK1 KO MEFs and AD model mice, suggesting the therapeutic potential of palmatine in neurodegenerative disease.

## 2. Results

### 2.1. Palmatine Induces Mitophagy in Mammalian Cells

To investigate the mitophagy-inducing activity of the natural alkaloid palmatine (Figure 1A), we analyzed cellular mitophagy activity upon palmatine treatment by using a previously established flow cytometry (FACS)-based mitophagy assay with pH-dependent mitochondria-targeted Keima (mt-Keima) [19,20]. We used CCCP, which is a well-known mitophagy inducer, as a control. FACS analysis upon treatment of the BEAS-2B human nontumorigenic lung epithelial cell line with palmatine (400 μM) revealed that palmatine treatment markedly increased the proportion of cells actively undergoing mitophagy (mitophagic cells (%)) with a similar level of CCCP treatment (Figure 1B). Quantitative analysis indicated that palmatine increased the number of mitophagic cells approximately 7.5-fold (Figure 1B). Confocal microscopy analysis upon treatment of the BEAS-2B cell line with palmatine (400 μM) showed a marked increase in the red puncta of mt-Keima, an indication of mitophagy [19] (Figure 1C). Palmatine treatment also reduced the fluorescence signal of mitochondrial YFP (mitoYFP) (Figure 1D), the levels of mitochondrial proteins, including Cox2, SDHB, and MFN2 (Figure 1E), and the level of mitochondrial DNA (Figure 1F). These results suggest that mitophagy activity increases upon palmatine treatment in BEAS-2B cells. Both an increase in mitophagic cells and a decrease in mitoYFP fluorescence were also observed upon palmatine treatment in HeLa-Parkin cells (Appendix A). An increase in mitophagic cells upon palmatine treatment was also observed in A549 human lung cancer cells (Appendix A). These results indicate that palmatine efficiently induces mitophagy in various human cells.

### 2.2. Palmatine Specifically Induces Mitophagy and Subsequent Mitochondrial Biogenesis

To further verify palmatine-induced mitophagy, dose- and time-dependent mitophagy induction upon palmatine treatment was examined. As shown in Figure 2A, significant mitophagy induction was observed after 100 μM palmatine treatment and 400 μM palmatine treatment resulted in the maximum degree of mitophagy induction. In addition, a time-dependent increase in the percentage of mitophagic cells was observed (Figure 2B). We then investigated whether palmatine specifically stimulates mitophagy. It has been shown that macroautophagy levels can be measured by analyzing the punctate structure formation of Keima protein localized in the cytoplasm and that mitochondrion-targeted mt-Keima can be used to detect mitophagy [8,21]. We found that palmatine treatment did not induce the formation of cytosolic Keima puncta, while starvation readily increased the number of Keima puncta (Figure 2C), suggesting that palmatine specifically induces mitophagy.

Mitophagy stimulation has been shown to induce subsequent mitochondrial biogenesis to maintain mitochondrial homeostasis [22]. Quantitative RT–PCR analysis revealed that palmatine treatment induced the expression levels of critical regulators of mitochondrial biogenesis, such as PGC-1α, NRF1, and TFAM, beginning 48 h later (Figure 2D–F), suggesting that palmatine-induced mitophagy subsequently induces mitochondrial biogenesis.

### 2.3. Palmatine Does Not Interfere with Mitochondrial Function

Mitochondrial inducers, such as CCCP, extensively employed in experimental settings, cannot be applied clinically due to their adverse effects on mitochondria [23]. Thus, we next examined whether palmatine interferes with mitochondrial function. The TMRM staining results revealed that the mitochondrial membrane potential (MMP) was not significantly reduced upon palmatine treatment for up to 6 h, while CCCP treatment readily suppressed MMP as early as 0.5 h (Figure 3A). In addition, an increase in mitochondrial ROS, another representative indicator of mitochondrial dysfunction, was not observed upon palmatine treatment for up to 24 h, while CCCP treatment increased mitochondrial ROS levels approximately 8.1-fold 24 h later (Figure 3B). Although the mean MitoSOX Red staining results were slightly increased upon palmatine treatment, these increases were not statistically significant. Moreover, while treatment with the antioxidant N-acetylcysteine (NAC) completely abolished CCCP-induced mitophagy, mitophagy induction by palmatine was not suppressed by NAC (Figure 3C,D), suggesting that palmatine-induced mitophagy is not dependent on mitochondrial ROS levels. These results suggest that palmatine-mediated mitophagy is not dependent on mitochondrial dysfunction.

### 2.4. Palmatine Alleviates Mitochondrial Dysfunction in PINK1-Knockout Mefs

Our results suggest that palmatine efficiently induces mitophagy but is not toxic to mitochondria. Thus, we next examined whether palmatine-induced mitophagy is able to reverse established mitochondrial dysfunction. To test this possibility, we examined the effect of palmatine treatment on PINK1 knockout (PINK1 KO) MEFs, a well-known mitochondrial dysfunction model accompanying various phenotypes, such as an increase in mitochondrial ROS and a decrease in the respiration rate [24,25]. Consistent with previous reports, compared with control wild-type MEFs, PINK1 KO MEFs showed increased mitochondrial ROS levels and decreased respiration (Figure 4A,B). Interestingly, palmatine treatment reduced the mitochondrial ROS level in PINK1 KO MEFs to wild-type MEF levels (Figure 4A). Moreover, palmatine treatment in PINK1 KO MEFs significantly improved mitochondrial respiration (Figure 4B). Basal respiration and maximal respiration improved by approximately 29% and 60%, respectively. These results suggest that palmatine alleviates mitochondrial dysfunction in PINK1 KO MEFs.

### 2.5. Palmatine Ameliorates Cognitive Deficits and Mitochondrial Dysfunction in an AD Mouse Model

To further verify the effect of palmatine in a more physiological setting, PS2APP AD model mice expressing mutant presenilin 2 (PS2 N1411) and mutant amyloid precursor protein (APP Swe) were treated with palmatine (10 mg/kg) daily for 4 weeks and cognitive function was assessed using the Morris water maze test (Figure 5A). Palmatine was administered intranasally to deliver it directly to the CNS. Throughout the training sessions conducted on days 2–6, it was evident that PS2APP mice exhibited a significantly higher latency in locating a submerged hidden platform compared to their wild-type control (Figure 5B), suggesting a notable impairment in their spatial learning ability. Notably, the administration of palmatine improved learning performance during the training session (Figure 5B). In addition, the results from the Morris water maze test on day 7 showed that the reduced time spent and distance traveled in the target quadrant by PS2APP mice with palmatine treatment were significantly mitigated to the level of wild-type mice (Figure 5C,D). These results suggest that the impaired spatial learning and memory ability of PS2APP mice was relieved upon palmatine administration. We subsequently isolated mitochondria from the hippocampal region in each group of mice and analyzed mitochondrial function. Importantly, the reduced mitochondrial membrane potential of PS2APP mice was reversed by palmatine administration to the level of wild-type mice (Figure 5E). The increased mitochondrial ROS levels and decreased ATP levels were also significantly reversed by palmatine administration (Figure 5F,G). In addition, western blot analysis revealed that the decreased mitochondrial biogenesis factors, including PGC-1α, NRF1, and TFAM, in the PS2APP mouse brain were significantly reversed by palmatine administration (Figure 5H–K), suggesting that palmatine-induced mitophagy also simultaneously stimulates mitochondrial biogenesis in mouse brain. The elevated levels of the inflammatory cytokines IL-1β, IL-6, and TNFα in the PS2APP mouse brain were also significantly decreased by palmatine administration (Figure 5H,L–N). These results suggest that palmatine alleviates cognitive impairments and mitochondrial dysfunction in an AD mouse model.

## 3. Discussion

Since the therapeutic effect of mitophagy stimulation on neurodegenerative diseases has been recently demonstrated, the identification of mitophagy inducers has become an important research topic. In this study, we found that the natural alkaloid palmatine induces mitophagy activity in various human cell lines, including BEAS-2B human lung epithelial cells, HeLa-Parkin cells, and A549 human lung cancer cells. While macroautophagy was not induced by palmatine treatment, palmatine simultaneously induced mitochondrial biogenesis following treatment. Interestingly, palmatine treatment did not significantly change the mitochondrial membrane potential or mitochondrial ROS levels. Moreover, unlike CCCP, treatment with the antioxidant NAC did not suppress palmatine-induced mitophagy, indicating that palmatine does not interfere with mitochondrial function to induce mitophagy. The toxicity of widely used mitophagy inducers, including CCCP and FCCP, restricts their clinical application [23]. For example, CCCP induces mitochondrial dysfunction and cell death by depolarizing the mitochondrial membrane potential [23]. Thus, our results suggest that palmatine is suitable for clinical application due to its low toxicity to mitochondria.

The effect of palmatine was verified in PINK1 KO MEFs. Palmatine treatment significantly reduced the elevated level of mitochondrial ROS and improved mitochondrial respiration in PINK1 MEFs, suggesting that palmatine is able to alleviate mitochondrial dysfunction. Numerous previous studies have demonstrated that mitochondrial dysfunction plays a pivotal role in the pathogenesis of various neurodegenerative diseases, including Alzheimer’s disease [26,27,28]. Various mitochondrial abnormalities, such as altered mitochondrial morphology, increased mitochondrial DNA mutation, and decreased activity of key mitochondrial enzymes, have been observed in both AD patients and AD animal models [29]. In particular, the early-stage observation of impaired mitochondrial function in AD models suggests that mitochondrial dysfunction is involved in the pathogenesis of AD [30,31,32]. In this context, Swerdlow and Khan proposed a mitochondrial cascade hypothesis that age-related mitochondrial dysfunction plays a central role in the pathogenesis of sporadic AD [33,34]. Consistent with this hypothesis, subsequent studies have shown that mitochondrial dysfunction influences the accumulation of phosphorylated Tau and Aβ, the representative pathological factors of AD [35,36,37]. Moreover, previous studies have also shown that Aβ and pathological Tau abrogate mitochondrial function [38,39,40,41]. These results suggest that mitochondrial dysfunction and major pathological factors such as Aβ and pathological Tau form a vicious cycle during AD development [27,42]. Recent studies have revealed reduced mitophagy activity in various AD models [43,44,45,46] and AD patients [47,48], suggesting that impaired mitophagy may act as a driver of mitochondrial dysfunction and the pathogenesis of AD [27]. Supporting this notion, the stimulation of mitophagy has shown a therapeutic effect in AD mouse models [47,49,50,51,52]. Consistent with these results, our study showed that palmatine administration significantly ameliorated both impaired cognitive function and mitochondrial dysfunction in PS2APP AD model mice. This further strengthens the concept that mitophagy stimulation is a promising strategy for AD treatment by disrupting the vicious cycle of mitochondrial dysfunction in the development of AD. Whether palmatine also inhibits the accumulation of Aβ and pathological Tau needs to be further investigated.

In addition, considering the pivotal role of PINK1 in Parkinson’s disease, it is conceivable that palmatine treatment could also prove advantageous in PD models. Further exploration is needed to determine whether palmatine administration exhibits therapeutic effects in PD models. While palmatine has a long history of use in traditional herbal medicine, DNA toxicity and impacts on metabolic enzymes in the liver have also been reported [12]. Therefore, there is a vital need to mitigate this toxicity for clinical purposes while preserving palmatine activity in promoting mitophagy through structural modifications. It is also necessary to improve the mitophagy activity of palmatine through chemical optimization. In addition, it is critical to elucidate the molecular mechanisms that underlie palmatine-induced mitophagy to facilitate its clinical application. Further studies are needed to help determine the specific mitophagy pathway by which palmatine is responsible. While our study focused on male mice due to the availability of PS2APP mice, it is necessary to verify whether palmatine exhibits a similar effect in female mice of the same age, as a previous study indicated that Aβ deposition in the cerebrum progresses slightly faster in female mice [53]. Further research is also warranted to comprehensively assess in vivo toxicity, including toxicity to various organs and body weight, to ensure the safe and effective clinical application of palmatine as a potential therapy for Alzheimer’s disease.

Taken together, our study identified palmatine as a potential mitophagy inducer with low toxicity. Although palmatine has been implicated in various biological functions, to our knowledge, its role in mitophagy regulation has not been investigated. Nevertheless, while the exact molecular mechanism should be further investigated, our results suggest that mitophagy stimulation could be a potential strategy for the treatment of AD.

## 4. Materials and Methods

### 4.1. Cell Lines, Plasmids and Treatments

BEAS-2B, A549, HeLa-Parkin and MEF cells were maintained in DMEM containing 10% fetal bovine serum (FBS; JR Scientific Inc., Woodland, CA, USA). PINK1 KO MEFs were kindly provided by Dr. Jongkyeong Chung (Seoul National University, Seoul, Republic of Korea) [54]. Cell lines stably expressing mt-Keima or Keima were generated via infection with a lentivirus produced by using a pLVX-mtKeima lentiviral construct [23].

CCCP was purchased from Sigma–Aldrich (St. Louis, MO, USA). Palmatine (P0245) was purchased from LKT Laboratories (St. Paul, MN, USA).

### 4.2. Measurement of Mitophagy and Autophagy Levels

To measure mitophagy activity through a flow cytometry-based assay, mt-Keima-expressing cells were treated with palmatine and mt-Keima fluorescence was examined using an LSR Fortessa flow cytometer (BD Biosciences, Franklin Lakes, NJ, USA) equipped with 405 nm and 561 nm lasers at the Neuroscience Translational Research Solution Center (Busan, Republic of Korea) as described previously [20]. The percentage of cells undergoing mitophagy (mitophagic cells (%)) was determined by gating cells exhibiting a high ratio of emission at 561 nm/405 nm excitation. To distinguish between high and low ratios of emission at 561 nm/405 nm excitation, we used untreated HeLa-mt-Keima cells exhibiting low mitophagy activity as the standard for a low ratio. The experiments were independently repeated three times and the results are presented as the mean ± SD.

To measure mitophagy levels in cells and tissue samples, mt-Keima fluorescence was detected using a Zeiss LSM 700 confocal microscope equipped with a C-Apochromat 40×/1.20 W Korr M27 lens at the Neuroscience Translational Research Solution Center (Busan, Republic of Korea). mt-Keima fluorescence was imaged using two sequential excitation lasers (458 nm and 561 nm) and a 595–700 nm emission bandwidth. Quantitation of mitophagy levels based on mt-Keima confocal images was performed using Zeiss Zen software (Zen 3.0 SR) as described previously [32,49]. We depicted the depicted the mt-Keima fluorescence signal from the 458 nm excitation wavelength in green and the signal from excitation by the 561 nm laser in red. The mitophagy level (mitophagy (%)) was defined as the number of pixels with a high red/green ratio divided by the total number of pixels. To quantify the mitophagy level in cells, the experiment was independently repeated three times and at least five images per sample were analyzed in each experiment.

To measure autophagy levels in cells, Keima fluorescence was analyzed, and the level of autophagy (autophagy (%)) was determined in the same manner as the level of mitophagy described above. In all confocal microscopy analyses, all imaging parameters remained constant and only the gain level was adjusted to avoid the saturation of any pixel. All mitophagy and autophagy measurement results are presented as the mean ± SD.

### 4.3. Confocal Microscopy of Fluorescent Mitochondrial Marker

To analyze the level of mitochondria upon ATL001 treatment, cells expressing fluorescent markers for mitochondria (mitoYFP) were examined using a Zeiss LSM 700 confocal microscope at the Neuroscience Translational Research Solution Center. To determine the fluorescence intensities of organelle-specific markers, at least five images per sample were analyzed. The experiment was independently repeated three times and the results are presented as the mean ± SD.

### 4.4. Western Blot Analysis

Cells were lysed in RIPA buffer and subjected to western blot analysis as described previously. Anti-Cox2 (ab198286), anti-SDHB (ab14714), anti-MFN2 (ab56889), anti-PGC-1α (ab191838), anti-NRF1 (ab175932), and anti-TFAM (ab272885) antibodies were purchased from Abcam (Cambridge, UK). An anti-actin (SC-47778) antibody was purchased from Santa Cruz (Dallas, TX, USA). Anti-IL-1β (#12242), anti-IL-6 (#12912,), and anti- TNFα (#11948) antibodies were purchased from Cell Signaling Technology (Danvers MA, USA). Band intensities were quantified using densitometry and ImageJ software (Image J 1.52k) (NIH, Bethesda, MD, USA).

### 4.5. Measurement of Mitochondrial DNA Content

To determine the mitochondrial DNA content, total genomic DNA including mitochondrial DNA was isolated as described previously [15]. The mitochondrial DNA content was quantified using quantitative real-time PCR with a probe that specifically recognizes the mitochondrial D-loop. The amount of 18S rDNA used to quantify nuclear DNA, was simultaneously determined in the same reaction and was used as the internal control. Quantitative real-time PCR was performed as described previously [15]. The sequences of the primers used in this study were as follows: mitochondrial DNA (D-loop), 5′-GATTTGGGTACCACCCAAGTATTG-3′ and 5′-GTACAATATTCATGGTGGCTGGCA-3′; and nuclear DNA (18S rDNA), 5′-TAGAGGGACAAGTGGCGTTC-3′ and 5′-CGCTGAGCCAGTCAGTGT-3′.

### 4.6. Quantitative RT–PCR

For quantitative real-time PCR analysis, total RNA was isolated using an easy-BLUE™ Total RNA Extraction Kit (iNtRON Biotechnology, Seongnam, Republic of Korea) following the manufacturer’s instructions and cDNA was synthesized using TOPscript™ RT DryMIX (Enzynomics, Daejeon, Republic of Korea). Quantitative real-time PCR was performed in triplicate in a SYBR Green PCR Master Mix (Enzynomics) using an ABI Prism 7500 Real-Time PCR System (Thermo Fisher Scientific, Waltham, MA, USA). β-Actin was used as an internal control for all samples, with normalization of gene-specific mRNA levels to the β-actin RNA level. The level of each mRNA was determined using the 2^−ΔΔCΤ^-threshold cycle method.

The primers used were as follows: PGC-1α forward primer (5′-ATGTGTCGCCTTCTTGCTCT-3′) and PGC-1α reverse primer (5′-ATCTACTGCCTGGGGACCTT-3′); TFAM forward primer 5′-GTCCATAGGCACCGTATTGC-3′ and TFAM reverse primer 5′-CCCATGCTGGAAAAACACTT-3′; NRF1 forward primer, 5′-CAACAGGGAAGAAACGGAAA-3′ and NRF1 reverse primer 5′-GCACCACATTCTCCAAAGGT-3′; and Actin forward primer 5′-CATGTACGTTGCTATCCAGGC-3′ and Actin reverse primer 5′-CTCCTTAATGTCACGCACGAT-3′.

### 4.7. Measurement of Mitochondrial Membrane Potential and Mitochondrial ROS

Mitochondrial membrane potential was measured using tetramethylrhodamine methyl ester (TMRM, 100 nM) (Invitrogen, Carlsbad, CA, USA) and mitochondrial ROS levels were measured using MitoSOX Red (5 μM) (Invitrogen, Carlsbad, CA, USA) as described previously [55]. The TMRM fluorescence intensity and MitoSOX Red fluorescence intensity were analyzed via flow cytometry using an LSR Fortessa cytometer.

### 4.8. Mitochondrial Respiration Analysis

Cellular respiration rates were measured in a 24-well plate using an XF24 flux analyzer (Seahorse Bioscience Inc. North Billerica, MA, USA) as described previously [55]. The oxygen consumption rate was measured under basal conditions followed by the sequential addition of oligomycin (0.5 μM), carbonyl cyanide *p*-trifluoromethoxyphenylhydrazone (FCCP, 1 μM), and rotenone (1 μM)/antimycin A (1 μM) to assess basal respiration, proton leakage, maximal respiration, nonmitochondrial respiration, and ATP production. The oxygen consumption parameters were normalized to the number of cells. The experiment was repeated five times and the results are presented as the mean ± SD.

### 4.9. Animal Experiments

PS2APP model mice (C57BL/6-Tg(NSE-hPS2*N141I);Tg(NSE-hAPP Swe)) were obtained from the National Institute of Food and Drug Safety Evaluation (NIFDS, Cheongju, Korea), and all procedures were performed in accordance with a protocol approved by the Dong-A Institutional Animal Care and Use Committee (DIACUC-22-19).

For the analysis of the therapeutic effect of palmatine on the AD mouse model, ten-month-old PS2APP male mice were intranasally administered 10 mg/kg palmatine dissolved in solution (5% DMSO, 10% Tween 80) daily for four weeks. All animal experiments were performed in a blinded fashion.

### 4.10. Morris Water Maze

The Morris water maze test was performed to analyze long-term learning and spatial memory as described previously [56,57]. A circular target platform (10 cm diameter) was immersed in a pool (diameter of 120 cm, depth of 50 cm) and a high-contrast cue was attached to the inside of the pool near the platform above the water surface. The test was conducted every 24 h for 7 consecutive days. Before starting the main experiment, all mice were allowed to swim in the presence of the cue and a 90-s visible platform trial was performed on day 1 to allow adaptation to water. On days 2–6, the mice were placed in the water with their heads facing the wall of the pool. In the hidden platform trials, the water was made opaque and the mice were placed in the pool four times in different quadrants for 5 consecutive days. On day 7, the platform was removed from the pool and the probe trial was performed for 90 s. The swimming trajectories were video-recorded. The distance traveled and time spent in the quadrant containing the platform were measured using Smart software (Smart 2.0) (Panlab, Barcelona, Spain). The experimenter was blinded to the treatment of the animals and the data analysis.

### 4.11. Hippocampal Mitochondria Isolation

Mouse hippocampal mitochondria were isolated following a previously established method with modifications [32,58]. Briefly, mice were euthanized via CO_2_ asphyxiation, mouse brains were quickly removed, washed with ice-cold PBS, and the hippocampus was isolated from the left hemisphere. The right hemispheres were saved for subsequent analysis. The final mitochondrial pellet was suspended in isolation buffer 3 containing 215 mM mannitol, 75 mM sucrose, 0.1% BSA, and 20 mM HEPES (pH 7.2) to yield a final protein concentration of approximately 1 mg/mL and immediately stored on ice. Twenty micrograms of mitochondria were used for TMRM staining, MitoSOX Red staining, and an ATP assay. For the analysis of ATP levels in the mouse hippocampus, isolated mitochondria (20 μg) were resuspended in ATP extraction buffer and analyzed using the same ATP assay system. ATP content was analyzed in four to six mice per group and the results are presented as the mean ± SD.

## Figures and Tables

**Figure 1 ijms-24-16542-f001:**
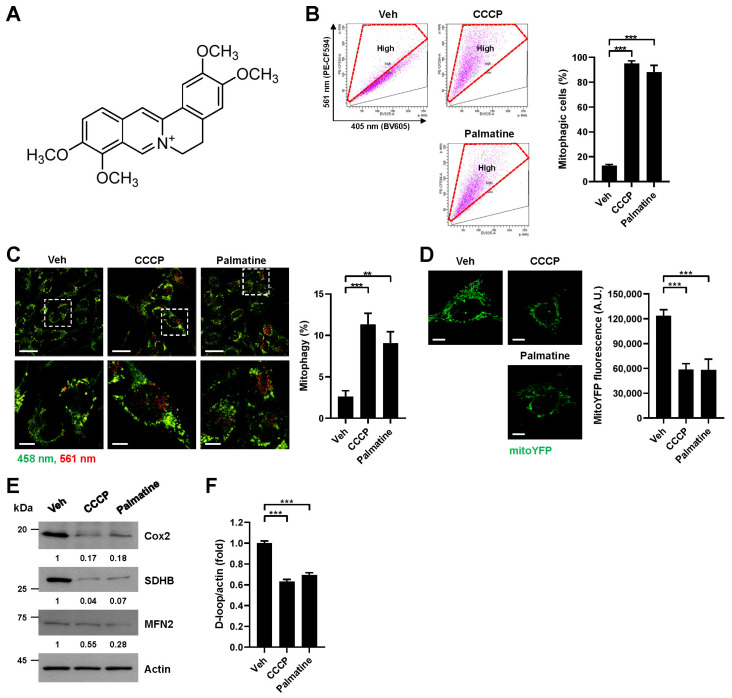
Verification of mitophagy induction by palmatine. (**A**) Chemical structure of palmatine. (**B**–**F**) BEAS-2B cells expressing mt-Keima (**B**,**C**), BEAS-2B cells expressing mitoYFP (**D**) or BEAS-2B cells (**E,F**) were treated with carbonyl cyanide m-chlorophenyl hydrazone (CCCP) (10 μM) or palmatine (400 μM) for 24 h. Mitophagy levels were analyzed by flow cytometry (**B**). The results from repeated experiments are shown as the mean ± SD. Mitophagy levels were analyzed by confocal microscopy (**C**). The results from three biological replicates are shown as the mean ± SD. Scale bar: 20 μm (upper). The boxed regions are enlarged in the bottom panel. Scale bar: 10 μm (bottom). The fluorescence intensity of mitoYFP was analyzed by confocal microscopy (**D**). Quantified fluorescent intensities from three biological replicates with several images per biological repeat are shown on the right as the mean ± SD. Scale bar: 10 μm. Cell lysates were subjected to western blot analysis using the indicated antibodies (**E**). Numbers below the corresponding blot represent densitometry values normalized to actin. The intracellular mitochondrial DNA (mtDNA) content was analyzed by real-time PCR with primer sets targeting the mitochondrial DNA D-loop and actin as a control for nuclear DNA (**F**). The results from three biological replicates are shown as the mean ± SD. Significance was determined by one-way ANOVA with Šidák’s multiple-comparison test. ** *p* < 0.01; *** *p* < 0.001.

**Figure 2 ijms-24-16542-f002:**
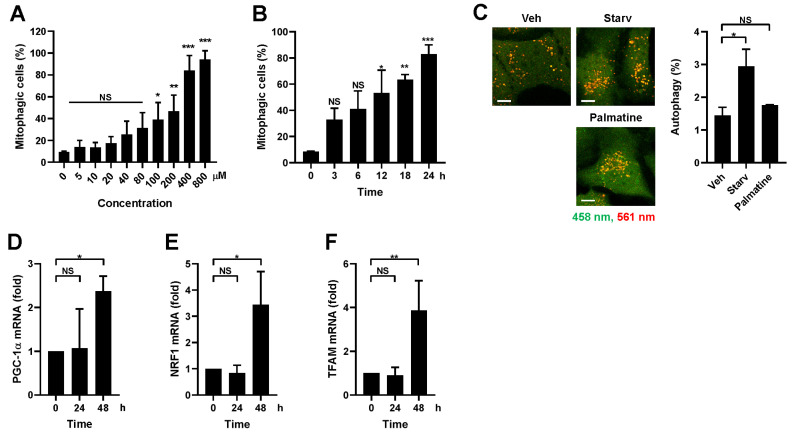
Analysis of mitophagy-specific induction and mitochondrial biogenesis upon palmatine treatment. (**A**) BEAS-2B cells expressing mt-Keima were treated with palmatine at the indicated concentration for 24 h and mitophagy levels were analyzed via flow cytometry. (**B**) BEAS-2B cells expressing mt-Keima were treated with palmatine (400 μM) for the indicated periods and mitophagy levels were analyzed via flow cytometry. The results from three biological replicates are shown as the mean ± SD. (**C**) BEAS-2B cells expressing Keima were starved for 3 h or treated with palmatine (400 μM) for 24 h and autophagy levels were analyzed using confocal microscopy. Scale bar: 10 μm. (**D**–**F**) BEAS-2B cells were treated with palmatine (400 μM) for 24 h. Cells were harvested at the indicated time points and the mRNA levels of PGC-1α (**D**), NRF1 (**E**), and TFAM (**F**) were measured via quantitative real-time PCR. The results from three biological replicates are shown as the mean ± SD. Significance was determined by one-way ANOVA with Šidák’s multiple-comparison test. * *p* < 0.05; ** *p* < 0.01; *** *p* < 0.001. NS, not significant.

**Figure 3 ijms-24-16542-f003:**
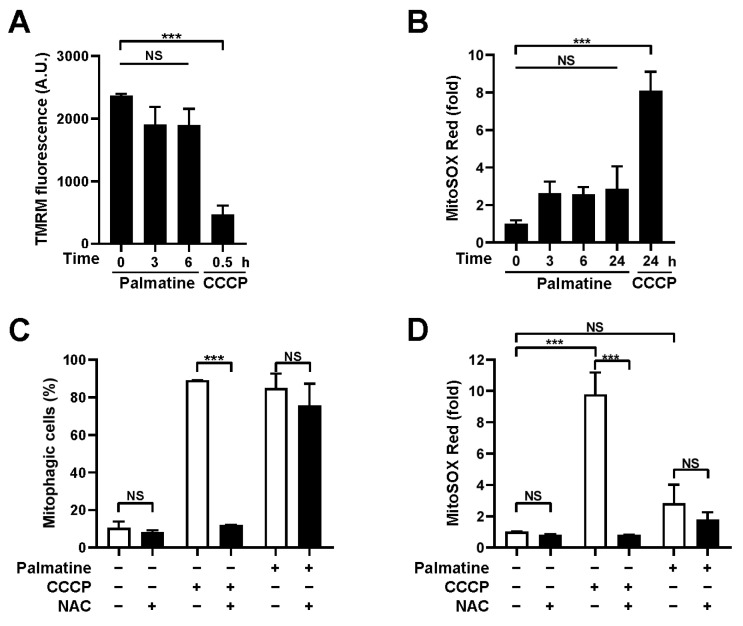
Effect of palmatine on mitochondrial function. (**A**,**B**) BEAS-2B cells were treated with CCCP (10 μM) or palmatine (400 μM) for the indicated periods. The mitochondrial membrane potential was assessed by TMRM staining (**A**) and mitochondrial ROS levels were determined by MitoSOX Red staining (**B**). The results from three biological replicates are shown as the mean ± SD. (**C**) BEAS-2B cells expressing mt-Keima were treated with CCCP (10 μM) or palmatine (400 μM) alone or cotreated with N-acetylcysteine (NAC, 2 mM) for 24 h and mitophagy levels were analyzed by flow cytometry. The results from repeated experiments are shown as the mean ± SD. (**D**) BEAS-2B cells were treated with CCCP (10 μM) or palmatine (400 μM) alone or cotreated with NAC (2 mM) for 24 h and mitochondrial ROS levels were determined by MitoSOX Red staining. The results from three biological replicates are shown as the mean ± SD. Significance was determined by one-way ANOVA (**A**,**B**) or two-way ANOVA (**C**,**D**) with Šidák’s multiple-comparison test. *** *p* < 0.001. NS, not significant.

**Figure 4 ijms-24-16542-f004:**
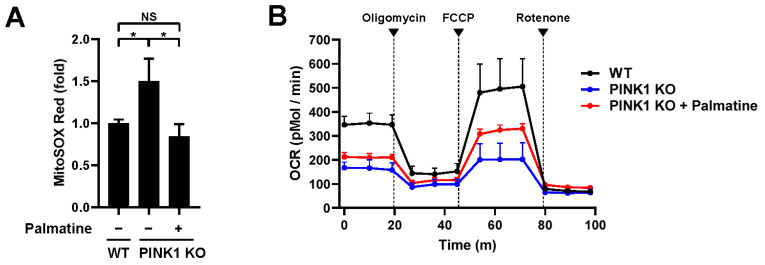
Effect of palmatine on PINK1 KO MEFs. (**A**,**B**) Wild-type (WT) MEFs and PINK1 KO MEFs were treated with palmatine (400 μM) for 24 h and cultured for an additional 4 days. Mitochondrial ROS levels were assessed by MitoSOX Red staining (**A**). The results from three biological replicates are shown as the mean ± SD. Mitochondrial respiration was analyzed by an XF-24 analyzer with five samples per group (**B**). * *p* < 0.05. NS, not significant.

**Figure 5 ijms-24-16542-f005:**
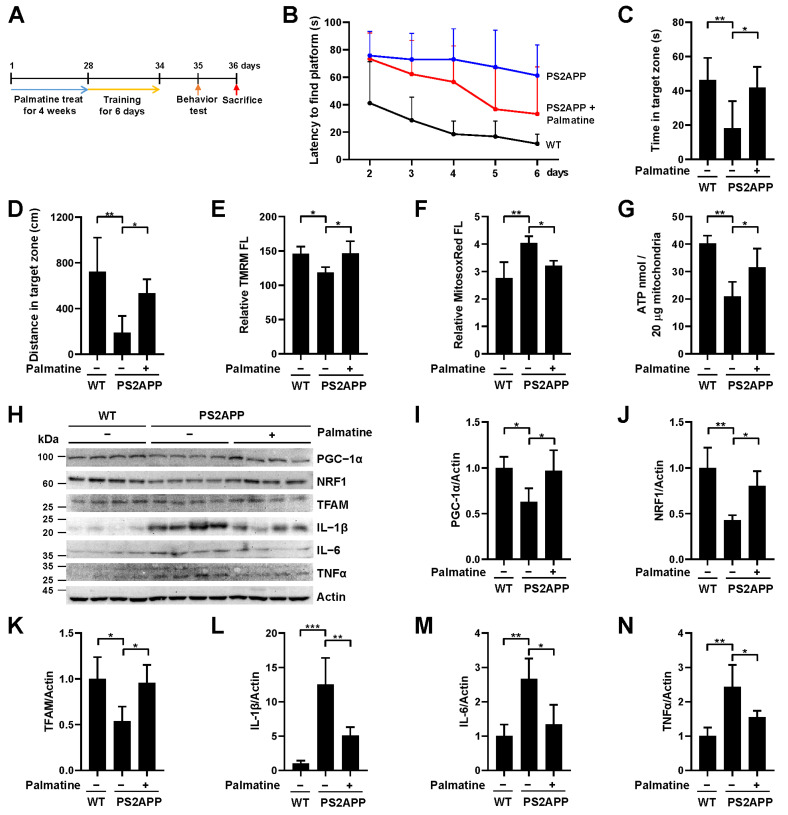
Effect of palmatine on cognitive decline and mitochondrial dysfunction in an AD mouse model. (**A**) Schematic diagram of the animal experiments. (**B**–**D**) PS2APP male mice were treated with palmatine (10 mg/kg) or vehicle (−) via intranasal administration daily for 4 weeks (n = 6 per group). The average latency to find the hidden platform in the target quadrant over 5 consecutive days (**B**). Time spent in the target zone (**C**) and the distance traveled in the target quadrant (**D**) in the Morris water maze test during the probe trial session on the 7th day. (**E**–**G**) PS2APP mice in (**B**–**D**) were sacrificed after the Morris water maze test and mitochondria were isolated from the hippocampus. Using isolated mitochondria, mitochondrial membrane potential (**E**), mitochondrial superoxide levels (**F**), and ATP levels (**G**) were analyzed (n = 4 per group). (**H**–**N**) PS2APP mice in (**B**,**D**) were sacrificed after the Morris water maze test and brain lysates were subjected to western blot analysis using the indicated antibodies (n = 4 per group). Quantified protein levels from four mice (**I**–**N**). The results are shown as the mean ± SD. Significance was determined by one-way ANOVA with Šidák’s multiple-comparison test. * *p* < 0.05; ** *p* < 0.01; *** *p* < 0.001.

## Data Availability

The data that support the findings of this study are available from the corresponding author, J.Y., upon reasonable request.

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
