# Peer review of "The Natural Alkaloid Palmatine Selectively Induces Mitophagy and Restores Mitochondrial Function in an Alzheimer’s Disease Mouse Model"

_ijms, 2023, doi:10.3390/ijms242216542_

Round 1

Reviewer 1 Report

Comments and Suggestions for Authors

In this paper, the authors describe the effects of palmatine, a natural alkaloid, on the activation of mitophagy, a crucial process for maintaining mitochondrial functions, in different human cell lines and in a mouse model of Alzheimer's disease.  They show that palmatine is able to induce mitophagy without interference with mitochondrial activity (ATP production, membrane potential etc.). The methods and results are well described. 

I have only some suggestions/requests regarding the animal treatment and results:

1) in the Morris water maze test, normally the mouse models of AD show a strong impairment in the learning activity. Is it possible to show also the learning curve of the behavioral test? Have the authors noted an improvement in this parameter?

2) the authors indicate in the introduction the anti-inflammatory effect of palmatine. Is it possible to determine the effect of palmatine on the expression of IL1 or IL6 in the brain of mice? These interleukins are involved in the development of cerebral inflammation in AD, and are involved in the mitochondrial dysfunction in this condition.

3) Have the authors analyzed the mitophagy specific induction and mitochondrial biogenesis upon palmatine treatment in the brain of mice?

4) I think it is necessary to be very cautious in stating that "palmatine is suitable for clinical application according to its low toxicity to mitochondria". The authors show only one data regarding the mitochondrial toxicity of palmatine on the cellular line, but no data are shown after treatment on the AD animal model. For instance, have they analysed the macroscopic effect on the animal liver? Have they measured the density of pycnotic/apoptotic cells in the liver? Have they weighed the animals every day before the treatment? Is there an effect of treatment on the body weight?  The clinical application of a new therapy requires many studies at different levels. 

In the legend of Figure 1 there are some mistakes in the assignment of letters.

Reviewer 2 Report

Comments and Suggestions for Authors

In a manuscript submitted for review, the Authors described the natural alkaloid palmatine selectively induces mitophagy and restores mitochondrial function in an Alzheimer's disease mouse model. I find the topic of the manuscript interesting and "up to date" and the whole work is thoughtful. The Authors put a lot of work into preparing this interesting work.

my comment/suggest:

1.     unfortunately, there is no information on how the animals were killed. Was it decapitation or was CO2 used?

2.     there is no information as to why only the left hippocampus was collected for testing, and there is no information as to why male mice were used in the study. what about females?

3.     there is no information on how many animals were used in the experiment.
